# Exploring the photosensitizing potential of Nanoliposome Loaded Improved Toluidine Blue O (NLITBO) Against *Streptococcus mutans*: An in-vitro feasibility study

**Swagatika Panda**[1]\*, **Lipsa Rout**[2], **Neeta Mohanty**[1], **Anurag Satpathy**[3], **Bhabani Sankar Satapathy**[4], **Shakti Rath**[5], **Divya Gopinath**[6,7]\*

1 Department of Oral Pathology and Microbiology, Institute of Dental Sciences, Siksha'O'Anusandhan Deemed to be University, Bhubaneswar, Odisha, India, 2 Institute of Dental Sciences, Siksha'O'Anusandhan Deemed to be University. Bhubaneswar, Odisha, India, 3 Department of Periodontics and Implantology, Institute of Dental Sciences, Siksha'O'Anusandhan Deemed to be University, Bhubaneswar, Odisha, India, 4 Gitam School of Pharmacy, Gitam (Deemed to be) University, Hyderabad, Telangana, India, 5 Department of Microbiology & Research, Institute of Dental Sciences, Siksha'O'Anusandhan Deemed to be University, Bhubaneswar, Odisha, India, 6 Basic Medical and Dental Sciences Dept, College of Dentistry, Ajman University, Ajman, United Arab Emirates, 7 Centre of Medical and Bio-Allied Health Sciences and Research, Ajman University, Ajman, United Arab Emirates

\* swagatikapanda@soa.ac.in (SP); d.gopinath@ajman.ac.ae (DG)

**Data Availability Statement:** All relevant data are within the manuscript

## Abstract

### Background

*Streptococcus mutans* is a major contributor to dental caries due to its ability to produce acid and survive in biofilms. Microbial resistance towards common antimicrobial agents like chlorhexidine and triclosan has shifted the research towards antimicrobial Photodynamic therapy (PDT). In this context, Toluidine Blue O (TBO) is being explored for its photosensitizing properties against *Streptococcus mutans*. There is a huge variation in the effective concentration of TBO among the current studies owing to the differences in source of and delivery system TBO as well as the time, power and energy densities of light.

### Objective

The primary objectives of this study are to encapsulate improved Toluidine Blue O (ITBO) in nanoliposomes (NLITBO), characterize it, and evaluate its antibacterial photosensitizing potential against *Streptococcus mutans* suspensions in vitro.

### Method

ITBO was synthesised as per Indian patent (number -543908). NLITBO was prepared using the thin-film hydration method. Dynamic light scattering experiment determined the vesicle size, polydispersity index (PDI), and zeta potential. Surface features were characterized by Scanning and Transmission Electron microscopy. ITBO release from NLITBO was assessed using the extrapolation method. The antibacterial activity of the NLITBO was determined by evaluating the zone of inhibition (ZOI) in the *Streptococcus mutans* culture

**Funding:** The author(s) received no specific funding for this work.

**Competing interests:** The authors have declared that no competing interests exist.

and comparing with 2% chlorhexidine gluconate. The minimum inhibitory concentration (MIC) of NLITBO as a photosensitizer with red light (wavelength 650nm, power density 0.1 W/cm$^2$, energy density 9–9.1 J/ cm$^2$, 90seconds time) was evaluated against *Streptococcus mutans* cells by colorimetric method in 96 well plate.

## Results

Percentage drug loading, loading efficiency, yield percentage, vesicle size, PDI, Zeta potential of NLTBO was reported as 9.3±0.4%, 84.4±7.6%, 73.5%, 123.52 nm, 0.57, -39.54mV respectively. Clusters of uni-lamellar nanovesicles with smooth non-perforated surfaces were observed in SEM and TEM. The size of the vesicle was within 100 nm. At 24 hours, a cumulative 79.81% of ITBO was released from NLITBO. Mean ZOI and MIC of NLITBO (1 μg /ml) were found to be 0.7±0.2 mm, 0.6μg/ml respectively.

## Conclusion

We have synthesized and encapsulated improved Toluidine Blue O (ITBO) in nanoliposomes (NLITBO) and thoroughly characterized the formulation. The antibacterial efficacy of NLITBO without light was demonstrated by ZOI which is similar to 2% chlorhexidine gluconate. MIC of NLITBO as a photosensitiser along with the optimal light parameter was also proposed in this study. These findings suggested that NLITBO could serve as an effective alternative to conventional antibacterial treatments in managing *Streptococcus mutans* rich biofilms. It can have potential pharmaceutical application in oral health care.

## 1. Introduction

Dental caries, the most common infectious oral cavity disease, is a biofilm-mediated progressive disease that results in tooth cavitation [1]. *Streptococcus mutans* is the major cariogenic species in the dental biofilm. Tooth surface adhesion, biofilm formation, acidogenicity through glycolysis, aciduricity, and glucosyltransferase-mediated extracellular polysaccharide synthesis are the cariogenic attributes of *Streptococcus mutans* [2]. Biofilms are complex, structurally organized microbial colonies and extracellular polymeric substances (EPS), a biochemical mixture of polysaccharides, proteins, glycopeptides nucleic acids and lipids [3]. EPS restricts the entry of antimicrobials, and host immune response components, thereby offering protection to microorganisms. The primary challenge for the conventional antibacterial agents such as chlorhexidine and triclosan is penetrating the biofilm. Biofilm matrix absorbs the antibacterial agents, limiting their diffusion and availability within the biofilm interior [4, 5]. Furthermore, the small pores of EPS impede the passage of larger molecules [6], necessitating significantly higher antimicrobial doses. Enzymatic inactivation of penetrated antibacterial agents within the biofilm further complicates its effectiveness [7]. Further, the antibacterial effectiveness is challenged by the dormant growth phase of bacteria inside the biofilm [8]. Once inside the biofilm, antibacterial agents primarily exert their effects by damaging cell walls or interfering with metabolic processes intracellularly [9]. This triggers the emergence of antimicrobial resistance through various mechanisms, including mutation, acquisition of exogenous resistance genes, horizontal transfer of resistance genes, and efflux pump activity [8, 10]. Additionally, the mosaic structure of biofilms generates gradients of oxygen, nutrients, pH, and redox potential, leading to heterogeneity in growth rate and metabolic activity across different biofilm layers,

which demand high specificity of antibacterials [11]. Besides microbial resistance, allergic reactions to chlorhexidine [12], ability to form calculus by being incorporated into the dental plaque [13], and increased microleakage [13]. These challenges encourage research on exploring alternatives to the existing antimicrobials. Antimicrobial photodynamic therapy (aPDT) has appeared as a non-invasive alternative to modifying cariogenic biofilm [14, 15] where light of appropriate wavelength activates photosensitizers, thus releasing singlet oxygen. This singlet oxygen eventually causes tissue deterioration and cell death through necrosis or apoptosis.

The antibacterial effect of Toluidine Blue O (TBO) as a photosensitizer has been explored in a few studies [16–19], particularly against gram-positive bacteria like *Streptococcus mutans* due to their cationic and hydrophilic nature. However, there is wide variation in the effective antibacterial concentration of TBO against *Streptococcus mutans*, which owes to inconsistency in the purity of TBO, delivery system of TBO and light parameters. Adequate penetration of the TBO into the biofilm is a noteworthy challenge for antibacterial efficacy, which in turn is dependent upon the purity and delivery system of TBO. Moreover, poor stability and limited bioavailability further limit the photosensitising properties of TBO. To optimize the photosensitising properties of TBO in harnessing the bacterial reduction outcome, we have attempted to minimise the above limitations of TBO-mediated aPDT by synthesising a purer form of TBO (ITBO). Nanoliposomes have been investigated for their biofilm penetrating potential, cell membrane-fusogenicity and biodegradability [10] which simultaneously improves the biocompatibility of the drug-loaded in it. Therefore, further, we attempted to load the ITBO in the core of the nanoliposome (NLITBO). The objective of the present research was to test the antibacterial efficacy of NLITBO against planktonic suspension of clinical strains of *Streptococcus mutans*. The antibacterial activity of NLITBO, when supported by such experiments, could be applied to caries and various other diseases where *Streptococcus mutans* is a primary component of the biofilm. The pharmaceutical industry could harness this technology to develop novel antimicrobial therapies against resistant biofilms, eventually improving patient outcomes and reducing healthcare costs associated with chronic infections.

## 2. Methodology

### 2.1. Synthesising Improved Toluidine blue O (ITBO)

ITBO was synthesised by the method as suggested in the Indian patent (patent number 543908 dated 01.07.2024 ) [20].

### 2.2. Preparation of NLITBO

To develop stable ITBO loaded nanoliposme (NLITBO), we selected plant based phospholipid, soya-L-α-lecithin (SL) (Sigma-Aldrich, CAS number 8002-43-5)as the primary lipid component [21, 22]. Cholesterol (CHL) (Sigma-Aldrich, CAS number 57-88-5) was used to stabilize the lipid structure [23]. This ITBO loaded nanoliposomes were developed using a modified thin film hydration method [24].

We prepared three formulations, each containing 15 mg of ITBO, with the ratio of SL to CHL increased progressively from 1:1, 1:3, and 1:5 and labelled as NLITBO 1, NLITBO 2, and NLITBO 3, respectively. SL and CHL were dissolved in 10 ml chloroform in a 250 ml round bottom flask. To this mixture, butylated hydroxyl toluene (2% w/v) (Sigma-Aldrich, CAS number 128-37-0) was added as an antioxidant to prevent oxidation of lipids. 15 mg of ITBO was added into the mixture and was shaken vigorously. The prepared mixture was then subjected to gentle rotation and evaporation of the solvent in a rotary vacuum evaporator (Rotavap, PBU-6, Superfit, Mumbai, India), connected with a water bath at 40°C. Evaporation of the solvent formed a thin film along the inner wall of the round bottom flask. The flask was

then kept in a desiccator overnight at room temperature to remove organic solvent residues left in the thin film. On the second day, the thin film was hydrated with phosphate buffer saline (PBS) (Himedia, ML023), pH 7.4, for 1 hour at 130–150 rpm rotation. After hydration, the mixture underwent sonication in a bath-type sonicator to reduce the size of liposomes to the nanoscale range. Post-sonication, the formulation was stored at 4˚C overnight. On the third day, the sample was centrifuged at 15,000 rpm for 45 minutes at 4˚C. The supernatant was discarded, and the sediments were collected and stored at –20˚C overnight. Finally, the pre-cooled samples were dried in a laboratory lyophilizer for 8–10 hours to obtain the dry powdered product.

## 2.3. Physico-chemical characterization of NLITBO

**2.3.1. Percentage yield, percentage drug loading and loading efficiency.** Lyophilized NLITBOs were weighed after each production batch, and the percentage yield was calculated as % Yield = (Amount of NLITBOs obtained after lyophilization / Total amount of components used in the formulation batch) × 100. Briefly, the amount of lyophilized NLITBOs was dissolved in ethanol and water in a 7:3 ratio. The sample was then vortexed for 2 min, sonicated for 5 min and centrifuged at 12,000 rpm. The supernatant was examined at 631.0 nm using a UV/VIS spectrophotometer (Beckman, Fullerton, CA, USA). The following formula was calculated for the percentage drug loading and loading efficiency in NLITBOs.

$$\text{Drug loading}(\%) = (\text{Amount of TBO in NLITBOs}/\text{Amount of NLITBOs obtained}) \times 100$$

$$\text{Drug loading efficiency}(\%) = (\text{Practical TBO loading}/\text{Theoretical TBO loading}) \times 100$$

**2.3.2. Determination of average vesicle size, polydispersity index (PDI) and Zeta potential.** To determine the average hydrodynamic size, PDI and zeta potential of NLTBOs, measurements were taken using a a Zeta sizer (Nano ZS 90, Malvern Instruments Ltd., UK) at 25 ± 0.5˚C. Lyophilized NLITBOs were diluted with deionized water and sonicatedfor 5 minutes. At 25 ± 0.5˚C, the sample was observed using a Zeta sizer.

**2.3.3. Determination of surface features.** The surface morphology and interior structure of the NLITBO nanovesicles were investigated using Scanning electron microscope (SEM), and Transmission electron microscope (TEM), respectively. The lyophilized samples were spread out over a stub on a carbon tape. The sample was then vacuum dried, followed by gold coating over the surface, and was observed under SEM (JSM 6100; JEOL, Tokyo, Japan). NLITBO was suspended in deionized water and put onto the carbon-coated copper mesh, which was then air-dried. The dried sample was then analysed under TEM (Tecnai G2 Spirit BioTwin FP 5018/40, FEI, Hillsboro, USA).

**2.3.4. ITBO release assessment in vitro.** 10 mg of lyophilized NLITBOs was suspended in phosphate-buffered saline (PBS, pH of 7.4) and was placed in a dialysis bag (Hi-media dialysis membrane-60, Mumbai, India). Two open ends of the dialysis bag were tied together using cotton thread, and the assembly was placed in a beaker filled with 50 millilitres of PBS. The beaker was placed over a magnetic stirrer having rotation speed at 300 rpm, and at fixed intervals. 1 millilitre of the release medium was removed with simultaneous replenishment of fresh PBS. The samples were analysed using a UV-visible spectrophotometer at a fixed wavelength of 631.0 nm. The amount of ITBO released from NLITBOs was determined by the extrapolation method from the standard calibration curve.

**2.3.5. Bacterial strain and culture condition.** Clinical strains of *Streptococcus mutans*, isolated from patients with caries as described previously [25], were cultured in blood agar

(Hi-media, M073) and maintained in the laboratory. The isolation and identification of the bacteria followed the standard microbiological procedure, ensuring the authenticity and clinical relevance of the strains. The strains were cultured in Muller Hinton agar media (Hi-media, M173) for experimentation. Bacterial Inoculum was prepared by inoculating *Streptococcus mutans* culture in Muller Hinton broth (Hi-media, M391) and incubated until the culture reached the logarithmic growth phase.

**2.3.6. Zone of inhibition (ZOI).**   *In vitro*, the antibacterial activity of NLITBO was determined by the agar well diffusion method. 0.5 McFarland standard culture ($1.5 \times 10^8$ colony forming unit/ ml) of *Streptococcus mutans* was prepared in Muller-Hinton agar media. 100 μL inoculum of *Streptococcus mutans* was spread onto Muller Hinton agar plates using a sterile cotton swab and allowed to dry, and wells were made with a sterile borer in the inoculated agar plates. 1 microgram of lyophilised NLITBO was dissolved in 1 ml of distilled water. 100 μL volumes of each test concentration were propelled into the culture plate of *Streptococcus mutans*. Then, the plate was incubated for 24 h at 37˚C. 10% aqueous Dimethyl sulfoxide (DMSO) (Emplura, SA1S70054) was used as the negative control; 2% chlorhexidine (ZodentaCG-148) was used as the positive control. The antimicrobial activity of NLITBO was observed by recording the ZOI. The experiments were performed in triplicates, and the mean diameter of the ZOI with standard deviation was recorded.

**2.3.7. Minimum inhibitory concentration (MIC) determination.**   The MIC of NLITBO was determined by the colourimetric method in 96 well microtiter plates (Tarsons, polypropylene v-bottomed square well with 200 μl capacity) [26]. A stock solution of NLITBO in distilled water was made. The designing of the 96-well microplate was performed as follows: One column each was designed for positive and negative control. Seven test columns were composed of serial concentrations of NLITBO (0.9, 0.8, 0.7, 0.6, 0.5, 0.4, 0.3 μg/ ml respectively). The microbial culture was inoculated into Nutrient broth (Hi-media, M002). The inoculum density was adjusted to approximately 0.5 McFarland standard ($1–2 \times 10^8$ CFU/mL).100 μl of NLITBO solutions having concentrations ranging from 0.9 μg to 0.3 μg along with 100 μl of the prepared bacterial culture were added in each column of 96 well microtiter plate ensuring the final volume in each well is 200 μL. The plate was covered to prevent contamination and light and incubated at 37˚C for 10 minutes. After incubation, each well was exposed to red LED light with a wavelength of 650nm and energy density approximating 9.1 J/cm$^2$. At the same energy density, the experiment was repeated thrice with three different combinations of power density and time (0.1 W/cm$^2$ + 90s, 0.08 W/cm$^2$ + 110s, 0.06 W/cm$^2$ + 150s). The light was applied to each well using a round tip with an 8 mm diameter attached to the light-emitting device (Novo Duo, Novolase, India). 20 μL of TTC solution (0.2% w/v) was added to each well, including the positive and negative controls. The positive control was without NLITBO, and the negative control was without the bacteria. Wells other than the receiving wells were blocked from light by aluminium foil. The tip of the light device accurately fits the surface of each well, which is approximately 7 mm in diameter, so a uniform sphere of light is applied in the suspension mix. The plate was incubated overnight at room temperature to allow the reduction of TTC by metabolically active microorganisms. The wells were compared to determine the lowest concentration of TBO, which shows no or minimal colour change or formazan formation. The colour change was read visually, and a stereomicroscope (Motic, USA) was used to observe red formazan precipitates in the wells. The experiment was conducted in triplicate, and the mean concentration of TBO at which bacterial growth was inhibited was accepted as the MIC.

**2.3.8. Statistical tests.**   Mean of ZOI among four groups were compared by Analysis of Variance (ANOVA) to observe the differences. To assess the variability across repeated experiments we calculated mean and standard deviation of MIC of TBO.

## 3. Results

### 3.1. Fabrication of experimental NLITBOs and determination of percentage yield, percentage drug loading, and percentage loading efficiency

The percentage drug loading for NLITBO-2 was 9.3±0.4%, whereas 3.6±1.1% and 5.2±1.02% for NLITBO-1 and NLITBO-3, respectively. NLITBO-2 also showed higher loading efficiency (84.4±7.6%) with higher yield percentage (73.5%) than the other two formulations (Table 1). Based on these parameters, NLITBO-2 was found to be the most suitable and we selected it for further studies.

### 3.2. Size analysis and zeta potential of selected NLITBO

DLS data revealed that the selected formulation (NLITBO-2) was well within the desired nano-size range (Z-avg = 123.52 nm) (Fig 1A). The PDI value was found to be 0.57. Zeta potential of NLITBO-2 was reported to be -39.54mV (Fig 1B).

### 3.3. Surface and internal structure analysis

SEM and TEM results depicted the successful formulation of NLITBO within the desired nano-range. All formed nanovesicles were spherical with smooth surfaces (Fig 2A). Most NLTBO-2 had a homogeneous size distribution, and there was a strong correlation between the vesicle sizes and the average size found in the DLS data. TEM study depicted a clear formation of uni-lamellar vesicular structures (Fig 2B). Nanoliposome vesicles with a spherical form, undistorted surface, and darker interior were observed in the TEM image (Fig 2B). The vesicles had no leaks or perforations, and the lamellarity was unbroken. The size of the vesicle was determined to be within 100 nm, which too supported by SEM and DLS data.

### 3.4. In vitro drug release study

The in vitro drug release study demonstrated sustained release characteristics of the NLITBO-2 at pH 7.4. At 24 h, a cumulative 79.81% of ITBO was released from NLITBO-2 (Fig 3).

### 3.5. Zone of inhibition (ZOI)

Mean ZOI of the positive control, i.e 2% chlorhexidine gluconate, 1 µg/ml NLITBO, 0.5 µg/ml NLITBO, 0.25 µg/ml NLITBO, and negative control were 1.2±0.3mm, 0.7±0.2mm, 0.3 ±0.1mm, 0 mm, and 0 mm respectively. ANOVA test demonstrated no significant difference between the four groups (p>0.05).

### 3.6. MIC of NLITBO

We observed no colour change in the test well containing 0.6 µg /ml ±0.03) NLITBO, indicating clear inhibition of bacterial growth to a gradual increase in the colour intensity as the

**Table 1. Composition, percentage yield, loading and efficiency of three formulations.**

| Formulation code | CHL:SL (w/w) | ITBO Quantity (mg) | % yield | Practical % loading | % Loading efficiency |
|---|---|---|---|---|---|
| NLTBO -1 | 1:1 | 15 | 61.4 | 3.6±1.1 | 58.3 ±5.1 |
| NLTBO -2 | 1:3 | 15 | 73.5 | 9.3±0.4 | 84.4±7.6 |
| NLTBO -3 | 1:5 | 15 | 56.6 | 5.2±1.02 | 61.3 ±0.4 |

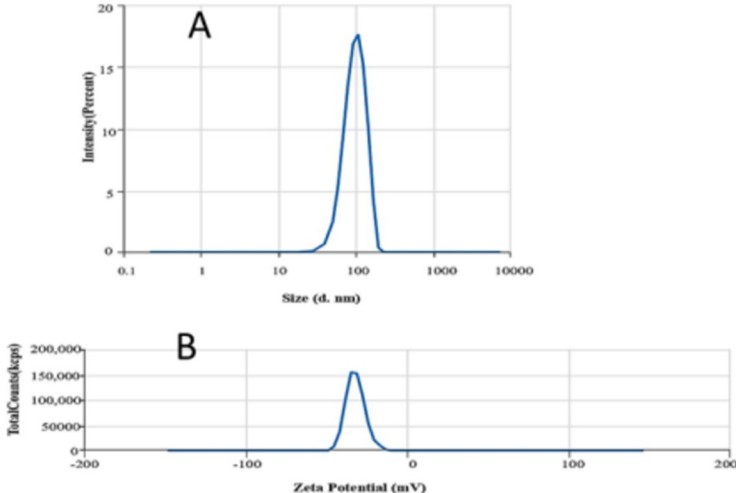

**Fig 1.** DLS demonstrating Size (A) and Zeta potential (B) of NLITBO.

concentration of NLITBO decreases (Fig 4). Therefore, we concluded that the MIC of NLITBO is 0.60 μg /ml ±0.03). Both positive and negative controls did not show any change.

## 4. Discussion

This experiment has explored the antibacterial efficacy of NLITBO against planktonic suspension of clinical strains of *Streptococcus mutans*, a significant cariogenic species. This study assessed the potential of NLITBO as a photosensitiser in aPDT, focusing on the advantage of nanoliposomes as a delivery system and the purer form of TBO (ITBO) as a photosensitiser.

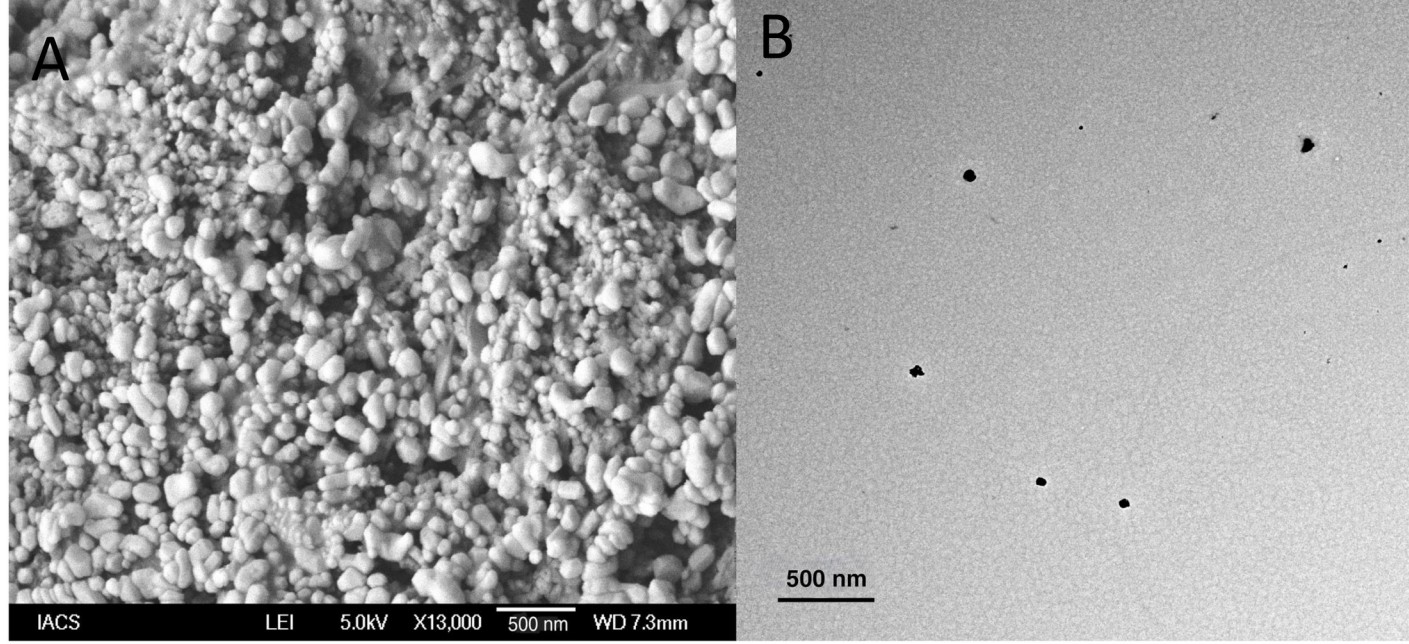

**Fig 2.** Surface structure in SEM (A) and interior structure in TEM (B) of NLITBO.

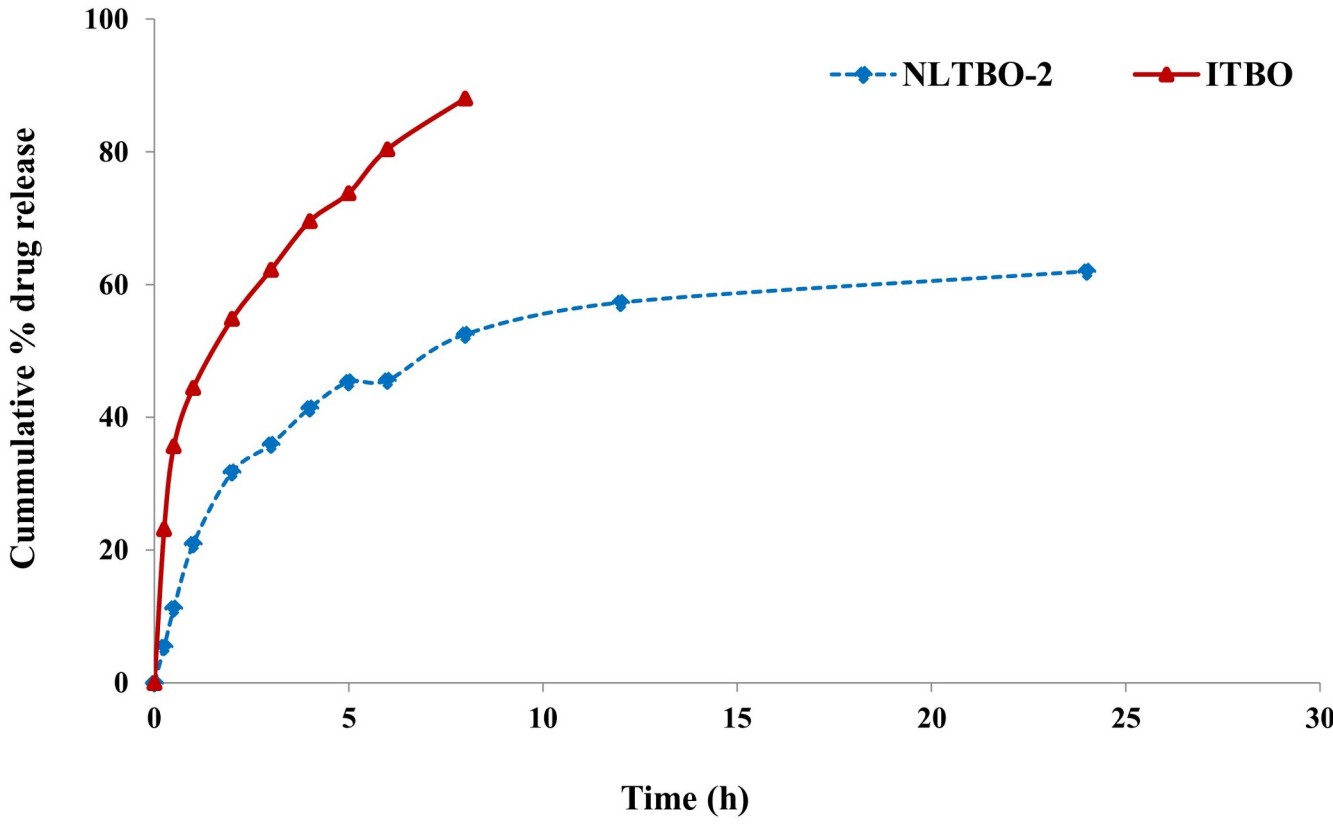

**Fig 3. In vitro release behaviour of NLITBO in PBS, pH 7.4.**

The results showed that NLITBO exhibited significant antibacterial activity against *Streptococcus mutans*. The ZOI produced by 1 µg/ml of NLITBO (0.7+0.2 mm) alone without PDT is not significantly different from the ZOI produced by 2% chlorhexidine (1.2+0.3 mm) (p

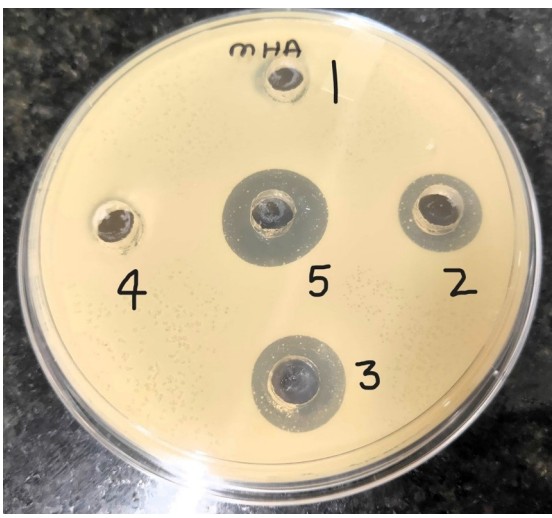

**Fig 4. Zone of Inhibition 1. NLITBO (0.25 µg /µl) - no zone; 2. NLITBO (0.5 µg /ml) - 0.3±0.1 mm.; 3. NLITBO (1 µg /µl) - 0.7±0.2 mm; 4. Negative Control - no zone; 5; Chlorhexidine gluconate 2% - 1.2 ± 0.3 mm.**

value>0.05) but promising enough to warrant further exploration of MIC of aPDT using NLITBO. This indicates that the antimicrobial effect of NLITBO is equivalent to chlorhexidine even without the light activation. This may be attributed to improved stability, improved cellular uptake, membrane disruption, and the synergistic effects of the nanoliposome carrier in NLITBO. Further experiment on antimicrobial effect of NLITBO activated by light can be explored.

The successful synthesis of ITBO and incorporation into the nanoliposome was well characterized by the appropriate composition ratio of SL and CHL (3:1), high percentage yield (73.5%), drug loading capacity (84.4±7.6), and surface and interior characteristics. Incorporating SL and CHL in 5: 1 ratio has been found to extend the stability of cinnamon oil for 96 hrs and enhance the anti-biofilm action against methicillin-resistant staphylococcus aureus (MRSA) by almost 10 times as compared to free cinnamon oil [27]. The lower value of PDI (0.57) indicates a homogenous and narrow distribution pattern of the formulation [28]. Zeta potential of -39.5mV classifies NLITBO as an anionic nanoliposome, which improves the stability in suspension. Higher similar charges between the nanovesicles create strong repulsive forces, keeping vesicles separated for longer. Stable nano formulations have been reported to have a surface charge beyond -30 mV to +30 mV [29, 30]. The drug release profile exhibited a controlled release pattern over the study period, which is beneficial for effective clinical application, reducing dose and dose-related side effects [31]. NLITBO showed antibacterial activity even without photoactivation, as observed by mean ZOI in the present study, which implied inherent antibacterial properties of NLITBO that can be further potentiated by red light application. This finding is negated by Shen et al [32] but supported by Nastri et al [33] who demonstrated the antibacterial effect of TBO at 10 µM concentration which was further potentiated by light application. The antibacterial activity of NLITBO without light may be attributed to the improved TBO and coating with anionic nanoliposome. NLITBO, owing to its zeta potential -39.5mV, is an anionic nanoliposome carrying cationic photosensitiser, ITBO. Anionic nanoliposomes may be less effective than cationic liposomes against gram-positive bacteria like *Streptococcus mutans* [34]. However, upon activation of the cationic photosensitiser, ITBO may penetrate the dental biofilm and *Streptococcus mutans* cell wall more effectively than cationic nanoliposome carrying cationic ITBO as it may not penetrate biofilm well due to the overall charge repulsion. Studies suggested that nanoliposomes loaded with cationic photosensitizers are designed for hydrophobic interaction with the negatively charged surface of gram-positive bacterial cells [35]. This allows for the diffusion of nanoliposome-loaded photosensitisers into *Streptococcus mutans*. The liposome-bacteria fusion was shown to enhance the diffusion of the drugs into *Pseudomonas aeruginosa* [36, 37]. This selective targeting enhances the specificity and efficacy of aPDT against gram-positive pathogens. Upon exposure to red light, the photosensitizer, ITBO loaded within the nanoliposomes, becomes excited [38]. It undergoes photochemical reactions, leading to the generation of reactive oxygen species (ROS), such as singlet oxygen and free radicals [39]. The generated ROS exert their antimicrobial effects by inducing oxidative damage to various cellular components of the gram-positive bacterial cells, eventually disrupting cellular functions and promoting cell death. This disruption facilitates the entry of ROS into the bacterial cytoplasm, enhancing their antimicrobial activity. Moreover, sustained release characteristics of the NLITBO-2, as found in this study, shall prove beneficial as there would be a consistent therapeutic effect even in the presence of saliva as the experiment was conducted at pH 7.4. A significant portion of ITBO (79.81%) was released from NLITBO-2 by the end of 24 hours. This attribute can be advantageous in a targeted and sustained approach during aPDT against gram-positive bacteria, providing a promising strategy for combating bacterial infections while minimising the risk of antimicrobial resistance and enhancing patient compliance.

This study has identified MIC of NLITBO against clinical strains of *Streptococcus mutans* suspension with red light (wavelength–650 nm, power density - 0.1mW/cm$^2$ for 90 seconds). Effective antimicrobial activity of TBO in *Streptococcus mutans* suspension was observed in the concentration range of 0.01 to 3.08 mg/ml [16–19, 40–43]. the effective concentration for anti-biofilm activity in *Streptococcus mutans* was 0.1mg/ml [19, 44–46] and 41.067 mg/ml. [45–49] Although the effective antibacterial concentration should vary according to the nature of bacterial culture, there is a huge difference in the effective concentration of TBO in studies on bacterial suspensions, too [16–19, 40–43]. This variation may be attributed to the difference in strains of s mutans, light parameters such as energy density, power density, incubation time and time of red-light exposure across studies. Energy densities reported in different studies ranged from 9J/cm2 to 180J/cm2 [19, 26, 33, 35–38], while power densities ranged from 100 mW to 500 mW. Duration of light exposure also varied from 30 seconds [50] to 12 minutes 50 seconds [18]. Except for one [9] rest of the studies have used the commercial strain. In general, clinical strains exhibit more genetic diversity, and the findings on clinical strains are applicable in clinical settings. Therefore, the present result can find a broader applicability. There are two clinical studies [27, 40] which evaluated the antimicrobial potential of TBO in children with early childhood caries but failed to significantly reduce the bacterial count of the saliva. This contradicting result could be due to the absence of an appropriate drug delivery system. Nano-liposomal formulation, an effective drug delivery system, is known to enhance the bioavailability and biodistribution of the loaded drug [51]. Afrasiabi et al. [52] have shown nanoliposomes' enhanced antimicrobial photodynamic therapy containing curcumin and doxycycline against *Aggregatibacter actinomycetemcomitans*. Therefore, we, for the first time, have emphasised the nanoliposome encapsulation of ITBO and demonstrated that NLITBO is an effective antibacterial agent against clinical strains of *Streptococcus mutans* under optimal red-light parameters (wavelength 650nm, power density 0.1 W, energy density 9–9.1 J, 90 seconds time) with a MIC of 0.6μg/ml. These conditions are effective, and the biocompatible concentration of NLITBO and light energy dosage are selected to target the 24-hour incubated bacterial suspension of *Streptococcus mutans*.

Firstly, our study demonstrated the bactericidal efficacy of NLITBO as a photosensitiser in aPDT. The observed MIC of 0.60 μg/mL for NLITBO indicates its potent bactericidal activity, positioning it as a promising candidate for caries prevention. The enhanced efficacy of NLITBO can be attributed to several factors. One plausible explanation is the improved bio-availability and targeted delivery of TBO facilitated by nanoliposomes. Nanoliposomes offer a stable and controlled release platform, allowing for prolonged exposure of TBO to bacterial cells, thus maximizing its antimicrobial effects and minimizing potential adverse effects on surrounding tissues. While the conventional antimicrobials often struggle to penetrate the extracellular polymeric substance barrier in the biofilm to effectively target bacteria, nanoparticles' (NP) ability to penetrate the biofilm can inhibit the biofilm formation [53]. NPs can exert the antibacterial activity through different mechanisms such as cell wall disruption, interaction with DNA/RNA or proteins, biofilm inhibition, or ROS generation [54]. There are three stages during the interaction of NP with biofilm. NP transported to the surface of the biofilm, gets attached to the biofilm followed by migration in the biofilm. Several factors such as physico-chemical properties of NPs and extracellular matrix influence how the individual phases take place [55]. Additionally, the nanoliposomes get attached to the cell wall of bacteria and enhance ITBO penetration into the bacterial cytoplasm, where it gets photoactivated and the nascent oxygen released causes bacterial death. Furthermore, the nanoliposome carrier system may protect ITBO from degradation and enhance its retention within the oral cavity, prolonging its therapeutic effects. Supporting our findings and the concept of enhancing the bioavailability of TBO through conjugation, Misba et al. [56] demonstrated that TBO-silver

nanoparticle conjugates exhibited superior antibiofilm activity against *Streptococcus mutans* compared to TBO alone. Similarly, Kishen et al. demonstrated that combining TBO with microbial efflux pump inhibitors led to successful biofilm inactivation [57].

Despite the promising findings of our study, several limitations should be considered. Firstly, our study was conducted in vitro, and further investigations are needed to assess the efficacy of NLITBO in vivo. Additionally, while our study focused on *Streptococcus mutans*, future research should explore its effects on other oral pathogens and its potential interactions with the oral microbiome. Furthermore, the safety profile of NLITBO, including potential cytotoxicity and adverse effects, warrants thorough evaluation before clinical translation. The sensitivity of planktonic bacteria to antimicrobial agents is far lower than biofilm-grown bacteria [58]. Therefore, further research may be directed to find out the biofilm inhibitory concentration and biofilm microbicidal concentration, which would effectively explore the efficacy, safety, and practical application of ITBO in the context of dental care to determine its role in preventing and managing infections associated with *Streptococcus mutans*.

In conclusion, our study demonstrates the enhanced antimicrobial efficacy of NLITBO and the MIC with optimum light parameters during aPDT against *Streptococcus mutans*. This research highlights that the purity and delivery system of ITBO as a photosensitiser may have clinical benefits when exploring caries preventive agents. Further research in dental biofilms and clinical trials are needed to validate these findings *in vivo*, assess safety profiles, and explore potential applications in clinical practice. Overall, NLITBO represents an innovative and promising approach to caries prevention, addressing critical challenges in antimicrobial therapy and paving the way for improved oral health outcomes.

## Author Contributions

**Conceptualization:** Swagatika Panda, Neeta Mohanty, Divya Gopinath.

**Data curation:** Swagatika Panda.

**Formal analysis:** Swagatika Panda.

**Investigation:** Swagatika Panda, Lipsa Rout, Anurag Satpathy, Bhabani Sankar Satapathy, Shakti Rath, Divya Gopinath.

**Methodology:** Swagatika Panda, Lipsa Rout, Neeta Mohanty, Anurag Satpathy, Bhabani Sankar Satapathy, Shakti Rath, Divya Gopinath.

**Software:** Divya Gopinath.

**Supervision:** Neeta Mohanty, Divya Gopinath.

**Validation:** Neeta Mohanty, Anurag Satpathy.

**Visualization:** Swagatika Panda.

**Writing – original draft:** Swagatika Panda, Divya Gopinath.

**Writing – review & editing:** Swagatika Panda, Lipsa Rout, Neeta Mohanty, Anurag Satpathy, Bhabani Sankar Satapathy, Shakti Rath, Divya Gopinath.

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
