## [Decision Letter · Decision Letter 0]

15 Jul 2024

PONE-D-24-19897Exploring the photosensitizing potential of nanoliposome loaded improved Toluidine Blue O (NLITBO) against Streptococcus mutans: An in-vitro feasibility studyPLOS ONE

Dear Dr. Gopinath,

Thank you for submitting your manuscript to PLOS ONE. After careful consideration, we feel that it has merit but does not fully meet PLOS ONE’s publication criteria as it currently stands. Therefore, we invite you to submit a revised version of the manuscript that addresses the points raised during the review process.

We look forward to receiving your revised manuscript.

Kind regards,

Geelsu Hwang, Ph.D.

Academic Editor

PLOS ONE

Journal Requirements:

3. We note that your Data Availability Statement is currently as follows: "All relevant data are within the manuscript and its Supporting Information files."

Reviewers' comments:

Reviewer's Responses to Questions

**Comments to the Author**

1. Is the manuscript technically sound, and do the data support the conclusions?

Reviewer #1: Yes

Reviewer #2: No

Reviewer #3: Yes

2. Has the statistical analysis been performed appropriately and rigorously? 

Reviewer #1: Yes

Reviewer #2: No

Reviewer #3: Yes

3. Have the authors made all data underlying the findings in their manuscript fully available?

Reviewer #1: Yes

Reviewer #2: No

Reviewer #3: Yes

4. Is the manuscript presented in an intelligible fashion and written in standard English?

Reviewer #1: Yes

Reviewer #2: No

Reviewer #3: Yes

5. Review Comments to the Author

Reviewer #1: Topic is innovative. Manuscript is well written. Statistical test is mentioned. All references from reference list are depicted in the manuscript. Conclusion is in conformity with the objectives of the study.

Reviewer #2: The manuscript is quite a preliminary study. Without proper data, its not ready to be published.

in Biofilms, there is always different bacterial species, so targeting alone, S. mutans is not the cure of the disease.

pls see attached file for details.

Reviewer #3: In this article, the authors prepared NLITBO, characterized its features, and evaluated the antibacterial activity against S.mutans. This would provide a new approach for preventing and treating dental caries.

I recommend following suggestion.

Introduction

-Line # 87-91: Add the latest reference; https://doi.org/10.1016/j.jtemb.2024.127448

-Line # 93-103: Add more detail about the biofilm and the effect of antibacterial agent on the biofilm. Compare with the other antibacterial agents and also add how the antimicrobial agent used in study are helpful in pharmaceutical industries.

Methodology

-Add new separate heading with detail of reagents/chemicals, instrumentation, media etc. used in present study.

-Line # 136-138: Add the reference.

-Line # 139: Reference?

-Add the references in each section of Methodology.

Results

-Make sure bacterial species name must be written in correct form in all text.

Discussion

-Line #: 282, 283: This is the part of result section

-Line # 286-289: Why ZOI produced by NLITBO were higher than those of chlorhexidine, give strong reason.

-Line # 292: Add reference.

-Line # 296, 297: Cite the proper reference.

-Line # 302: Reference?

-Line # 320: Reference?

-S. mutans, correct it with same pattern in all text.

-Add the explanation, how the NPs affect on the biofilm producing bacterial structure?

-Lines # 345-348: Unnecessary statement. Is this related to this study?

6. PLOS authors have the option to publish the peer review history of their article (what does this mean?). If published, this will include your full peer review and any attached files.

Reviewer #1: No

Reviewer #2: No

Reviewer #3: No

---

## [Author Response · Author response to Decision Letter 0]

24 Aug 2024

Reviewer # 1

 Topic is innovative. Manuscript is well written. Statistical test is mentioned. All references from reference list are depicted in the manuscript. Conclusion is in conformity with the objectives of the study.

Author response– Thank you for your encouraging words.

Reviewer #2: 

Reviewer comment 1: The manuscript is quite a preliminary study. Without proper data, its not ready to be published.in Biofilms, there is always different bacterial species, so targeting alone, S. mutans is not the cure of the disease.

Author Response - While we acknowledge the complexity of biofilm communities and the presence of multiple bacterial species, this study was designed as a feasibility study to explore the efficacy of NLITBO against the primary contributor of the cariogenic biofilm which is Streptococcus mutans. Future research will involve expanding the scope to include other bacterial species commonly found in the biofilm, thereby providing a more comprehensive understanding of the potential of NLITBO in targeting polymicrobial biofilms. Our findings lay the groundwork for such studies by demonstrating the photosensitizing potential of NLITBO against a critical cariogenic species.

Reviewer comment 2 The treatment of planktonic suspension will not address the real problem. Biofilm existence comprising of S mutans and many others in case of dental caries. 

Author Response – Yes, We agree with the reviewer and acknowledge that the treatment of planktonic suspensions will not address the real problem. However, our study aims to establish a foundational understanding of the antibacterial efficacy of NLITBO against S. mutans in planktonic form as a preliminary step. The promising result of this study may guide the next logical step to evaluate its effectiveness on dental biofilms. 

Reviewer comment 3 Why both SEM and TEM. One was sufficient?

Author Response: We studied the surface morphology of NLITBO by SEM and internal structure of NLITBO by TEM so that both surface and internal properties are evaluated

Reviewer comment 4 Please prepare a biofilm of dental bacteria in vitro and then test the formulation. In planktonic mode against one bacteria. Of no use

Author response: Thank you for the insight. We agree that testing NLITBO in dental biofilm would provide a clinically relevant evaluation of its antibacterial efficacy. However, due to the time constraint and complexity of establishing a reliable in vitro dental biofilm model, it may not be feasible to conduct such experiment. Planktonic solutions are often used as a first line evaluation of any novel antibacterial agent’s efficacy and the result guides further research. We appreciate your concern and looking forward to explore this avenue in further studies. 

Reviewer comment 5 :ATCC or accession number of S mutans

Author response: The S mutans strain was isolated clinically. The methods has been described in page 8 line 228-229

Reviewer comment 6 Statistical analysis?

Author response: Thank you for your valuable suggestions. We have elaborated more on the statistical analysis on page 10, line 278-280

Reviewer #3: 

In this article, the authors prepared NLITBO, characterized its features, and evaluated the antibacterial activity against S.mutans. This would provide a new approach for preventing and treating dental caries.

Author response– Thank you for your encouraging words.

Reviewer comment 1

Introduction

-Line # 87-91: Add the latest reference; https://doi.org/10.1016/j.jtemb.2024.127448

Author response: We have added the reference as suggested.

Reviewer comment 2

-Line # 93-103: Add more detail about the biofilm and the effect of antibacterial agent on the biofilm. Compare with the other antibacterial agents and also add how the antimicrobial agent used in study are helpful in pharmaceutical industries.

Author response: Thank you for the insightful comments. We have added the further details of biofilm including effect of antibacterial agents on the biofilm. We also compared with other antibacterial agents and added the implication of this antimicrobial agent, i.e NLITBO in pharmaceutical industry. 

Reviewer comment 3

Methodology

Add new separate heading with detail of reagents/chemicals, instrumentation, media etc. used in present study.

Author response: We have done the modification as suggested.

Reviewer comment 4

-Line # 136-138: Add the reference.

Author response: We have added the reference as suggested

Reviewer comment 5

-Line # 139: Reference

Author response: We have added the reference as suggested

Reviewer comment 6

-Add the references in each section of Methodology.

Author response: We have added the reference as suggested

Reviewer comment 7

Results

-Make sure bacterial species name must be written in correct form in all text.

Author response: We have done the modification as suggested.

Reviewer comment 8

Discussion

-Line #: 282, 283: This is the part of result section

Author response: We have done the modification as suggested.

Reviewer comment 9

-Line # 286-289: Why ZOI produced by NLITBO were higher than those of chlorhexidine, give strong reason.

Author response: Mean ZOI produced by NLITBO is not higher but not significantly different from mean ZOI produced by Chlorhexidine. As per your valuable suggestions I have substantiated with possible reasons in the discussion. 

Reviewer comment 10

-Line # 292: Add reference.

Author response: Those empty parentheses is to write the values as found in this study. We have added the details.

Reviewer comment 11

-Line # 296, 297: Cite the proper reference.

Author response: We have added the reference as suggested

Reviewer comment 12

-Line # 302: Reference?

Author response: We have added the reference as suggested

Reviewer comment 13

-Line # 320: Reference?

Author response: We have added the reference as suggested

Reviewer comment 14

-S. mutans, correct it with same pattern in all text.

Author response: We have done the modification as suggested.

Reviewer comment 15

-Add the explanation, how the NPs affect on the biofilm producing bacterial structure?

Author response: We have added the explanation as suggested

Reviewer comment 16

-Lines # 345-348: Unnecessary statement. Is this related to this study?

Author response :Thank you for raising the concern. We believe the sentences are not explanatory enough for which we have edited these lines again. These sentences are a critical analysis of the studies which were conducted to extablish the effectiveness of TBO as a photosensitizer. Only one study has used the clinical strain similar to our study. In general, clinical strains exhibit more genetic diversity and the findings on clinical strains are applicable in clinical settings. Therefore, we have highlighted the broader applicability of our results. Similarly contradicting our establishment of antibacterial efficacy of NLITBO, two studies could not prove clinical effectiveness of TBO in early childhood caries, which may be attributed to the use of laboratory grade TBO, lack of appropriate delivery method and/or lack of optimum light parameters. In this light we have discussed the importance of nanoliposome as a drug delivery system.

---

## [Decision Letter · Decision Letter 1]

3 Sep 2024

PONE-D-24-19897R1Exploring the photosensitizing potential of nanoliposome loaded improved Toluidine Blue O (NLITBO) against Streptococcus mutans: An in-vitro feasibility studyPLOS ONE

Dear Dr. Gopinath,

Thank you for submitting your manuscript to PLOS ONE. After careful consideration, we feel that it has merit but does not fully meet PLOS ONE’s publication criteria as it currently stands. Therefore, we invite you to submit a revised version of the manuscript that addresses the points raised during the review process.

As you see the comments from one of the reviewer, there are still a couple of concerns regarding your manuscript. Please address those comments. 

We look forward to receiving your revised manuscript.

Kind regards,

Geelsu Hwang, Ph.D.

Academic Editor

PLOS ONE

Journal Requirements:

Reviewers' comments:

Reviewer's Responses to Questions

**Comments to the Author**

1. If the authors have adequately addressed your comments raised in a previous round of review and you feel that this manuscript is now acceptable for publication, you may indicate that here to bypass the “Comments to the Author” section, enter your conflict of interest statement in the “Confidential to Editor” section, and submit your "Accept" recommendation.

Reviewer #2: (No Response)

Reviewer #3: All comments have been addressed

2. Is the manuscript technically sound, and do the data support the conclusions?

Reviewer #2: No

Reviewer #3: Yes

3. Has the statistical analysis been performed appropriately and rigorously? 

Reviewer #2: (No Response)

Reviewer #3: Yes

4. Have the authors made all data underlying the findings in their manuscript fully available?

Reviewer #2: No

Reviewer #3: Yes

5. Is the manuscript presented in an intelligible fashion and written in standard English?

Reviewer #2: No

Reviewer #3: Yes

6. Review Comments to the Author

Reviewer #2: The authors didnt address the comments.

The study is quite preliminary without any significant data.

Manuscript need to be proof read by Native English Speaker.

pls see attached file.

Reviewer #3: All the comments have been revised. I don't have anymore questions. The article can be accecpted for publication now.

7. PLOS authors have the option to publish the peer review history of their article (what does this mean?). If published, this will include your full peer review and any attached files.

Reviewer #2: No

Reviewer #3: No

---

## [Author Response · Author response to Decision Letter 1]

1 Oct 2024

Reviewer's Responses to Questions

1. Authors must know how to cite

Answer – Thank you so much. We have changed the parenthesis to square brackets as per the journal requirements.

2. This is not the appropriate way to write for methodology.(materials and instrumentations)

Answer – Thanks for raising the concern. We did this in the first revision as it was asked by one of the reviewers during the first revision. However, we revised it again as per your kind suggestion.

3. Too many typo mistakes in methodology 

Answer –Thank you for rightfully pointing out the typo mistakes. We have corrected those mistakes.

4. As mentioned earlier, though strains were isolated earlier, they must have accession or ATCC numbers.pls mention here.

Answer –Thank you for the insightful comment. We have not procured this strain from any commercial source, for which we are unable to provide th ATCC number. We have also not submitted the sequencing data to any database for which we do not have accession number either. However, we have identified the bacteria based upon colony morphology, Gram's stain and sugar fermentation tests.

5. Pls do perform an in vitro antibiofilm assay before drawing final conclusion.

Answer –Our study was designed as a preliminary investigation to explore the bactericidal efficacy of novel photosensitiser, i.e NLITBO against Streptococcus mutans in a planktonic state. As planktonic cultures provide a controlled environment to evaluate the direct impact of the photosensitizer, this approach allows for a clear understanding of the fundamental antibacterial activity before extending the research to more complex biofilm models and is a common practice in antimicrobial research. This approach allows for fine-tuning the photosensitizer’s parameters under simpler conditions, which is crucial before addressing the added complexities of biofilms. Evaluating the efficacy in biofilm would require further optimization of photosensitizer parameters such as light exposure, penetration, and dosage, which can be the extension of this work. In fact, there are several published studies which have followed a similar approach, starting with the evaluation of novel photosensitizers in planktonic cultures of Streptococcus mutans before progressing to biofilm studies, which are summarised below. 

Author/ year DOI Photosensitiser

Marco Aurelio Paschoal, 2014 https://doi.org/10.1089%2Fpho.2013.3656

Curcumin Vs Toluidine blue

Anna Carolina Borges Pereira a Costa, 2010 https://doi.org/10.1590/S1806-83242010000400007

Erythrosine Vs Rose bengal

Juliana P.M.L. Rolim, 2011 https://doi.org/10.1016/j.jphotobiol.2011.10.001

Methylene blue ,Toluidine blue ortho ,Malachite green ,Eosin, Erythrosine and Rose bengal 

Maria Ângela Lacerda Rangel Esper,2020 https://doi.org/10.14295/bds.2020.v23i2.1857

Erythrosine

Caroline C Tonon,2015 https://doi.org/10.5005/jp-journals-10024-1626

Curcumin 

Marco Aurelio Paschoal,2013 https://doi.org/10.1016/j.pdpdt.2013.02.002

Curcumin 

6. SEM and TEM images are not clear

Answer –Thank you for the suggestion. We have shot another photomicrograph with better resolution and Figure 2 has been updated.

7. Drug release graphs are not showing the control curve

Answer –Thank you very much. We ran the experiment along with the control and duly uploaded another graph (new Figure 3)

---

## [Decision Letter · Decision Letter 2]

9 Oct 2024

Exploring the photosensitizing potential of nanoliposome loaded improved Toluidine Blue O (NLITBO) against Streptococcus mutans: An in-vitro feasibility study

PONE-D-24-19897R2

Dear Dr. Gopinath,

We’re pleased to inform you that your manuscript has been judged scientifically suitable for publication and will be formally accepted for publication once it meets all outstanding technical requirements.

Kind regards,

Geelsu Hwang, Ph.D.

Academic Editor

PLOS ONE

Additional Editor Comments (optional):

Reviewers' comments:

Reviewer's Responses to Questions

**Comments to the Author**

1. If the authors have adequately addressed your comments raised in a previous round of review and you feel that this manuscript is now acceptable for publication, you may indicate that here to bypass the “Comments to the Author” section, enter your conflict of interest statement in the “Confidential to Editor” section, and submit your "Accept" recommendation.

Reviewer #3: All comments have been addressed

2. Is the manuscript technically sound, and do the data support the conclusions?

Reviewer #3: Yes

3. Has the statistical analysis been performed appropriately and rigorously? 

Reviewer #3: Yes

4. Have the authors made all data underlying the findings in their manuscript fully available?

Reviewer #3: Yes

5. Is the manuscript presented in an intelligible fashion and written in standard English?

Reviewer #3: Yes

6. Review Comments to the Author

Reviewer #3: All previous suggestions have been revised by the authors and there are no new comments. The paper can be accepted in its current state

7. PLOS authors have the option to publish the peer review history of their article (what does this mean?). If published, this will include your full peer review and any attached files.

Reviewer #3: No

---

## [Editor Report · Acceptance letter]

18 Oct 2024

PONE-D-24-19897R2 

PLOS ONE

Dear Dr. Gopinath, 

I'm pleased to inform you that your manuscript has been deemed suitable for publication in PLOS ONE. Congratulations! Your manuscript is now being handed over to our production team.

Kind regards, 

on behalf of

Dr. Geelsu Hwang 

Academic Editor

PLOS ONE